

# Salivary proteome of a Neotropical primate: potential roles in host defense and oral food perception

Fabiola Carolina Espinosa-Gómez[1,2,3,*], Eliel Ruíz-May[4,*], Juan Carlos Serio-Silva[2] and Colin A. Chapman[1,5,6,7]

[1] Department of Anthropology and McGill School of Environment, McGill University, Montreal, Quebec, Canada

[2] Red de Biología y Conservación de Vertebrados, Instituto de Ecología AC, Xalapa, Veracruz, México

[3] Facultad de Medicina Veterinaria y Zootecnia, Universidad Popular Autónoma del Estado de Puebla, Puebla, Puebla, México

[4] Red de Estudios Moleculares Avanzados, Instituto de Ecología AC, Xalapa, Veracruz, México

[5] Department of Anthropology, Center for the Advanced Study of Human Paleobiology, George Washington University, Washington DC, Washington DC, United States of America

[6] School of Life Sciences, University of KwaZulu-Natal, Scottsville, Pietermaritzburg, South Africa

[7] Shaanxi Key Laboratory for Animal Conservation, Northwest University, Xi'an, Xi'an, China

[*] These authors contributed equally to this work.

Corresponding author
Fabiola Carolina Espinosa-Gómez, fabiolacarolina.espinosa@upaep.mx

## ABSTRACT

**Background**. Saliva contains a very complex mixture of proteins for defense against microbiological pathogens and for oral food perception. Howler monkeys are Neotropical primates that can consume a mostly leaf diet. They are well known to thrive in highly disturbed habitats where they may cope with a diversity of dietary challenges and infection risks. We aimed to describe the salivary proteome of howlers to contribute to better understanding of their physiology.

**Methods**. We analyzed the salivary proteins of wild black howler monkeys (*Alouatta pigra*), by SDS-PAGE-1-D and Nano LC-MS/MS and categorized them by their function involved in host defense and oral food perception.

**Results**. Our proteomic analysis identified 156 proteins in howler saliva including a number of host defense peptides that are the first line of defense in mammals, such as defensin, cathelicidin, dermcidin, and lactotransferrin, and proteins with anti-bacterial, anti-fungal, and anti-viral capacity, such as IgA, IgG, IgM, BPI, salivary heat shock 70 kDa protein, beta-2-microbulin, and protein S-100. We also identified key proteins necessary for taste perception, including salivary carbonic anhydrase VI, cystatin D, IgA, and fatty acid-binding protein. Proteins to detect astringent foods were identifying, including four members of cystatins (A, B, C and D), lactoperoxidase, and histidine-rich proteins. No chitinase and amylase were identified as would be expected because howlers do not eat insects and little starch. These findings provide basic information to future studies in oral biology, ingestive physiology, and physiological ecology of mammals and non-human primates.

## INTRODUCTION

Saliva plays a crucial role handling both nutritious and toxic foods. Saliva maintains oral health by protecting the digestive tract, maintaining tooth strength, and providing antimicrobial activity against bacteria, viruses, and fungus (*Fábián et al., 2012*). Oral food perception is facilitated by salivary proteins (*Canon & Neyraud, 2017*; *Rodrigues et al., 2017*; *Fábián et al., 2015*), so that individuals may choose a nutritious diet and avoid harmful secondary metabolites or toxins found in some foods (*Lamy et al., 2017*). The function of saliva can vary with diet and its proteome may be influenced by pathogens (*Thamadilok et al., 2019*; *Karasov & Douglas, 2013*; *Lamy et al., 2010*; *Da Costa et al., 2008*). Thus, physical and chemical properties of saliva, specially its proteome, relates to the animal's health and their ability to feed safely in particular kinds of environment (*Lamy & Mau, 2012*).

Saliva plays an important role in defense against pathogens. Research on oral biology in humans and other mammal species has identified that salivary proteins and peptides displayed additive and synergistic anti-bacterial, antiviral, and anti-fungal functions (*Fábián et al., 2012*; *Wang, Peterson & Loring, 2014*). Salivary components allowing this include: immunoglobulins, chaperone 70 kDa heat shock proteins, lysozyme, amylase, histatins, proline-rich proteins (PRPs), peroxidases, mucins, bactericidal/permeability-increasing protein (BPI), BPI-like proteins, palate lung and nasal epithelial clone proteins (PLUNC), proteins S100, clusterin, defensin, and statherin (*Amerongen & Veerman, 2002*; *Amerongen, Bolscher & Veerman, 2004*; *Carneiro et al., 2012*; *Fábián et al., 2012*).

Food preferences also may correspond to the expression of some peptides and proteins in saliva, and the taste sensitivity for specific tastants (*Salles et al., 2010*; *Canon & Neyraud, 2017*). The gustatory sensation is the result of the interaction of water-soluble chemicals in the mouth with the taste buds, this interaction is mediated by ions, hormones and salivary proteins that function as tastant-binding proteins (*Scott, 2005*; *Fábián et al., 2015*; *Canon & Neyraud, 2017*). For instance, sweet-taste sensitivity in humans is related with higher levels of cystatins and lower levels of amylase in saliva (*Rodrigues et al., 2017*). Other salivary proteins allow fatty acid taste perception (*Mounayar et al., 2014*), such as carbonic anhydrase VI (CA-VI), cystatin SN, cystatin D, zinc-alpha-2-glycoprotein, fatty-acid binding protein, and proline-rich proteins (PRPs).

Other salivary proteins participate in the detection of astringency when they interact with plant secondary metabolites, such as polyphenols (*Horne, Hayes & Lawless, 2002*). This tactile sensation represents a warning cue discouraging the ingestion of foods with high concentrations of polyphenols (e.g., tannins), which are a plant defense against herbivory (*Freeland, 1991*). Salivary proteins precipitate polyphenols preventing its negative physiological effects (*Bennick, 2002*). It has been found in humans and some mammals; increased levels of some salivary proteins (e.g., basic PRPs, cystatin, statherin, histatins (histidine-rich proteins), mucins, amylase, IgA, glycoprotein 1 and 2) in response to astringent compounds that collaborate with the acceptance of food to make it less aversive and more palatable (*Canon & Neyraud, 2017*; *Martin, Kay & Torregrossa, 2019*; *Nayak & Carpenter, 2008*; *Ployon et al., 2018*; *Torregrossa et al., 2014*).

The diet of herbivorous represents a significant challenge because their foods contain different types and concentrations of plant secondary metabolites (*Foley, Iason & McArthur, 1999*). Among them, tannins are one of the most studied and they deter herbivore feeding through two principal effects. The first involves making foods unpalatable as they have an astringent and bitter taste (*Horne, Hayes & Lawless, 2002*). The second involves binding dietary proteins and digestive enzymes reducing protein and food digestibility (*Austin, Suchar & Hagerman, 1989*; *Martinez-Gonzalez et al., 2017*; *Moore et al., 2014*; *Robbins et al., 1987*). Therefore, salivary proteins are the first line of defense against dietary tannins (*Shimada, 2006*).

Howler monkeys (genus *Alouatta*) are the most folivorous New World primate and have the widest geographical distribution of any primate in the Americas. These monkeys do well in highly fragmented and perturbed landscapes (*Kowalewski et al., 2015*; *Chaves & Bicca-Marques, 2016*), which may mean that they select the right foods and have an effective host-defense system. Their diet is leaf-based or fruit-based according food availability (*Dias & Rangel-Negrín, 2015*). Their ability to eat fibrous (*Espinosa-Gómez et al., 2013*) tannin-rich leaves and toxic unripe fruits contribute to their adaptability (*Garber, Righini & Kowalewski, 2015*; *Milton, 1979*). Black howler monkeys (*Alouatta pigra*) can consume plants with high concentration of tannins (*Espinosa-Gómez et al., 2018*; *Righini, Garber & Rothman, 2017*) and these monkeys continuously secrete salivary proteins with tannin-binding affinity (*Espinosa-Gómez et al., 2018*). Their tannin-binding salivary proteins (TBSPs) might be PRPs, but this remains to be confirmed (*Espinosa-Gómez et al., 2018*).

Black howler monkeys face habitat loss and fragmentation, and thus deal with nutritional stress and a high risk of disease transmission (*Kowalewski et al., 2011*; *Chapman, Gillespie & Goldberg, 2005*; *Chapman et al., 2013*). The objectives of our study are to (i) identify the proteins of whole saliva of black howler monkeys (*Alouatta pigra*) by proteomic analysis, (ii) distinguish proteins/peptides related to oral food perception, and (iii) characterize proteins related with host-defense and antimicrobial properties.

## MATERIALS & METHODS

### Saliva samples

All research protocols reported here were reviewed and approved by the government of Mexico (SEMARNAT SGPA/DGVS/10426/14) and complied with the legal and ethical guidelines of the IUCN (1998), and of the Mexican authorities (*Diario Oficial de la Federación, 1999*). We used the saliva samples obtained by FCEG as part of a complementary research project to evaluate the relationship of dietary tannins and tannin-binding salivary proteins (FC Espinosa-Gómez, 2017, unpublished data).

Samples were obtained from 14 free-ranging black howler monkeys occupying four forest fragments near Balancán, Mexico (17°44′05″N; 91°30′17″W). This disturbed forest landscape lies within cattle pastures (*Pozo-Montuy et al., 2013*). Monkeys were darted and anaesthetized by a veterinarian with ketamine hydrochloride (8 mg/kg estimated body mass, Ketaset, Fort Dodge Animal Health, Iowa USA). Once monkeys were stabilized following sedation, the body weight was determined and the saliva flow was stimulated by

an intra-muscular administration of the parasympathomimetic compound pilocarpine-hydrochloride (0.5 mg/ body mass) (*Espinosa-Gómez et al., 2018*; *Da Costa et al., 2008*). The whole saliva was collected from the mouth of each monkey using a micropipette, placed in a tube, and immediately frozen in liquid nitrogen. All saliva samples were transported from the field to the Proteomic Lab at INECOL, AC in Xalapa, Veracruz, México in a cryogenic container and then stored in an ultra freezer at −80 °C until analysis.

## Saliva preparation and SDS-PAGE

At the lab, saliva aliquots were thawed, cells and debris were removed by centrifugation at 16,000 g for 10 min at 4 °C, and the supernatant was captured. We determined the salivary total protein concentration by the Bradford method (*Bradford, 1976*) using bovine serum albumin (BSA) as a standard. Absorbance was measured at 595 nm with a microtiter plate reader (SpectroMAX 340, Molecular Devices, Union City, CA, USA). We fractionated salivary proteins using 12% one-dimensional sodium dodecyl sulfate-polyacrylamide gel electrophoresis (SDS-PAGE) following Laemmli (*Laemmli, 1970*). The 1D-SDS PAGE (8 × 7.3 cm × 1.5 mm) was run with 30 μg of salivary total protein with SDS loading buffer 4:1 (Biorad, CA, USA). Molecular mass markers (Precision Plus Protein Dual Color Standards, BioRad 1610374, CA, USA) were run in each gel to calibrate the molecular masses of the salivary proteins. Protein bands were fixed with a mixture of 26% ethanol, 14% formaldehyde, and 60% water for 3 hr, followed by 3 hr in a mixture of 50% methanol and 12% acetic acid (Steck, Leuthard, & Bürk, 1980). We followed the procedures suggested by *Beeley et al. (1991)* to detect PRPs, which allows PRPs stain pink or pinkviolet. Briefly, gels were stained overnight with a 0.25% Coomassie brilliant blue R-250 solution (Biorad 1610400) in 40% (v/v) methanol and 10% (v/v) acetic acid. We de-stained the protein bands with several changes of 10% acetic acid.

## In-gel digestion proteins

The clear proteins bands observed in our protein gels provided sufficient clean samples for proteomic analysis using the Nano LC-MS/MS approach. Protein bands were manually removed from gels and cut into 13 different molecular weight ranges (bands a - m) by excising these regions with a sharp straight edge and then destained with 2.5 mM ammonium bicarbonate (NH4HCO3) in 50% acetonitrile (ACN), and then dehydrated with 100 μL of 100% ACN. Samples were then reduced with 20 μl of 10 mM DTT in 50 mM NH4HCO3 and incubated for 45 min at 56 °C. Subsequently, the samples were cooled to room temperature and proceeded with the alkylation by adding 20 μL of 100 mM iodoacetamide in 50 mM NH4HCO3, and incubating in the dark for 30 min. Then, the samples were washed with 100 μL of 100% ACN for 5 min, then with ¬100 μL of 5 mM NH4HCO3 for 5 min and then with 100 μL of 100% ACN for 5 min. Finally, samples were dried with CentriVap (Labconco Kansas, Missouri) for 5 min and rehydrated with 10 μL of digestion solution containing 12.5 ng/ μL mass spectrometry grade Trypsin Gold (Promega, Madison, WI, USA) in 5 mM NH4HCO3 and incubated in a water bath at 37 °C overnight. The reaction was stopped at −80 °C. The peptides were extracted with 30 μL of 50% acetonitrile with 5% formic acid by centrifugation at 1000× g for 30s and

desalted with ZipTip- µC18 tips (Merck Millipore, Darmstadt, Germany) and dried using a CentriVap (Labconco Kansas, Missouri, USA).

## Mass spectrometry (Nano LC-MS/MS analysis)

Suspended samples (5 µl of 0.1% formin acid) were injected into a nanoviper C18 trap column (3 µm, 75 µm X two cm, Dionex) at 3 µl min-1 flow rate and separated on an EASY spray C18 RSLC column (2 µm, 75 µm × 25 cm) with a flow rate of 300 nl min-1 connect to an UltiMate 3000 RSLC system (Dionex, Sunnyvale, CA) and interfaced with an Orbitrap FusoinTM TribidTM (Thermo-Fisher Scientific, San Jose, CA) mass spectrometer equipped with an "EASY Spray" nano ion source (Thermo-Fisher Scientific, San Jose, CA). For peptide separation, a chromatographic gradient using MS grade water (solvent A) and 0.1% formic acid in 90% acetonitrile (solvent B) for 30 min was set as followed: 10 min solvent A, 7–20% solvent B within 25 min, 20% solvent B for 15 min, 20–25% solvent B for 15 min, 25–95% solvent B for 20 min, and 8 min solvent A. The mass spectrometer was operated in positive ion mode with nanospray voltage set at 3.5 kV and source temperature at 280 °C. External calibrant included caffeine, Met-Arg-Phe-Ala (MRFA), and Ultramark 1621. The mass spectrometer was operated in a data-dependent mode to automatically switch between MS and MS/MS. The survey full-scan MS spectra were acquired in the Orbitrap analyzer, scanning of mass range was set to 350–1,500 m/z at resolution of 120,000 (FWHM) using an automatic gain control (AGC) setting to 4.0e5 ions, maximum injection time to 50 ms, dynamic exclusion 1 at 90S and 10 ppm mass tolerance. A top speed survey scan for 3s were selected for subsequent decision tree-based Orbitrap collision-induced dissociation (CID) or higher-energy collisional dissociation (HCD) fragmentation (*Swaney, McAlister & Coon, 2008*; *Frese et al., 2011*). The signal threshold for triggering an MS/MS event was set to 1.0e4 and the normalized collision energy was set to 35 and 30% for CID and HCD, respectively. The AGC of 3.0e4 and isolation window of 1.6 m/z was set for both fragmentations. Additional parameter for CID included activation Q was set to 0.25 ms and injection time to 50 ms. For HCD, first mass was set to 120 m/z and injection time to 100 ms. The settings for decision tree were as follows: for HCD fragmentation charge states 2 or 3 were scan in a range of 650–1,200 m/z, charge states 4 were scan in a range of 900–1,200 m/z, and charge states 5 were scan in a range of 950–1,200 m/z; for CID fragmentation charge states 3 were scan in a range of 650–1,200 m/z, charge state 4 were scan in a range of 300–900 m/z, and charge state 5 in scan range of 300–950 m/z. All data were acquired with Xcalibur 4.0.27.10 software (Thermo-Fisher Scientific).

## Database search and protein/peptide identification

Raw data were analyzed with Proteome Discoverer 2.1 (PD, Thermo Fisher Scientific Inc.) and subsequent searches were carried out using Mascot server (version 2.4.1, Matrix Science, Boston, MA) and SQUEST HT (*Eng, McCormack & Yates, 1994*). The search with both engines was conducted against *Homo sapiens, Macaca fascicularis, Macaca mulatta*, and the complete UniProt reference proteome (http://www.uniprot.org/). We included as parameters in the search: full-tryptic protease specificity, two missed cleavage allowed, static modifications covered carbamidomethylation of cysteine (+57.021 Da). Furthermore,

dynamic modifications included methionine oxidation (+15.995 Da) and deamidation in asparagine/glutamine (+0.984 Da). For the MS2 method, in which identification was performed at high resolution in the Orbitrap, precursor and fragment ion tolerances of $\pm 10$ ppm and $\pm 0.2$ Da were applied. Resulting peptide hits were filtered for maximum 1% FDR using the Target Decoy PSM validator. We considered a MASCOT score >20 for proteins identified with two or more peptides and MASCOT score >34 for proteins identified with one single peptide.

## Bioinformatic analysis

Proteins were screened for the predicted presence of N-terminal endoplasmic reticulum (ER) targeting signal peptide (SP) using the Signal P 4.1 program (http://www.cbs.dtu.dk/services/SignalP/, *Petersen et al., 2011*). In addition, we used the server Secretome P 2.0 to determine non-classical and leaderless protein secretion in proteins identified in the saliva of monkeys (http://www.cbs.dtu.dk/services/SecretomeP/, *Bendtsen et al., 2004*). The program MHMM server v. 2.0 were used for the prediction of transmembrane helices in salivary proteins (http://www.cbs.dtu.dk/services/TMHMM/). Proteins were classified base on GO ontology enrichment of biological processes using David ontology tool (*Sherman & Lempicki, 2009*) (https://david.ncifcrf.gov/). We used REVIGO web server (http://revigo.irb.hr/) with a median similarity for the visual representation of the clustering of biological processes.

## Search for proteins/peptides related with host-defense and taste sensitivity

To distinguish the salivary proteins related with taste sensitivity (beside with astringent detection in mouth), host defense, and antimicrobial properties (anti-bacterial, antiviral and anti-fungal), we carried out detailed scrutiny of the UniProt functional annotation (http://www.uniprot.org/) and also reviewed papers on salivary proteomics/peptidomics from humans and other animals that have identified proteins with specific functions on immunity and taste sensitivity of food. Most of the salivary proteins related with a specific function in an animal specie, has been identified in several others, which suggest that their function is conserved across species.

## RESULTS

### Salivary protein separation by SDS-PAGE

We observed similar salivary protein patterns on 1-D electrophoresis gels in all individuals. There were multiple bands (a–m) ranging from 10 to 250 kDa (Fig. 1), with the most intense protein bands being located at low molecular weight from 10–15 kDa (k, l, m). However, the intensity of the bands did vary, with the j band being more apparent in individuals P-M1 and P-F1, the band k was more intense in B-F2, and bands l and m displayed a darker and more significant area of staining in B-F2 and P-M1. We visualized a main protein band (j) with an apparent molecular mass between 22–30 kDa that displayed a pink staining, which might be PRP according to *Beeley et al. (1991)* and described in *Espinosa-Gómez et al. (2018)*.

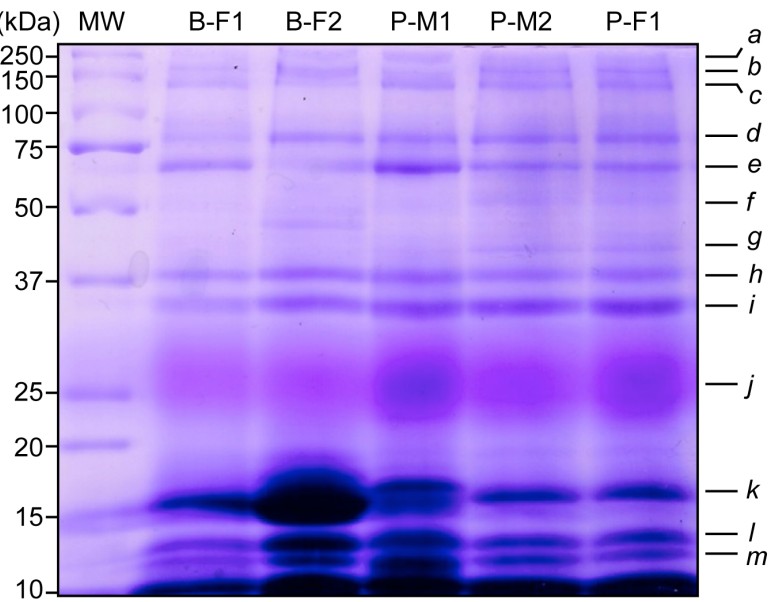

**Figure 1** **Representative SDS-PAGE-1-D of saliva proteins from five wild black howler monkeys. Protein bands were stained according to *Beeley et al. (1991)* to reveal potential PRPs.** We observed similar protein patterns from 10 to 250 kDa and identified 13 protein bands in all individuals ($N = 14$). Molecular weights (MW) of protein markers are shown in kDa on the left. B, Brisa group; P, Playon group; M, male; F, female.

## Identification of salivary proteins by Nano LC-MS/MS

We use proteomics to evaluate all 13-protein bands fractionated on SDS-PAGE-1D; we digested and subjected to LC MS-MS pools of the same protein band from all individuals, and 156 proteins were identified (Table S1). Among these, 55 were predicted with both signal peptide (SP) and transmembrane helices domains (TMHMM), including well-known secreted proteins, such as the Lactoperoxidase (P22079), Lactotransferrin (P02788), Serotransferrin (A5A6I6), the glycosylated Prosaposin (P07602), and the Histidine-rich glycoprotein (P04196). Besides, we were able to predict five non-secreted proteins with TMHMM (Fig. 2A). Using Secretome P 2.0 (http://www.cbs.dtu.dk/services/SecretomeP/, *Bendtsen et al., 2004*) we predicted ten proteins with a non-classical and leaderless secretion that include for example, Galectin-7 (P47929), Putative ubiquitin-conjugating enzyme E2 N-like (Q5JXB2), Putative ubiquitin-conjugating enzyme E2 N-like (Q5JXB2).

After gene ontology enrichment by David Bioinformatics Resources 6.8 (https://david.ncifcrf.gov/, *Huang, Sherman & Lempicki, 2009*) and clustering by REVIGO web server (http://revigo.irb.hr/, *Supek et al., 2011*), we obtained a tree map displaying key biological processes associated with howler monkey saliva, including negative regulation of endopeptidase activity, defense response to fungus, gluconeogenesis, protein folding, cytoskeleton organization, platelet degranulation, and epidermis development. Each of these major groups included several gene ontology (GO) groups (Fig. 2B). The most representative group corresponded to negative regulation of endopeptidase activity that clustered gene ontology, such as proteolysis (GO:0006508), protein stabilization

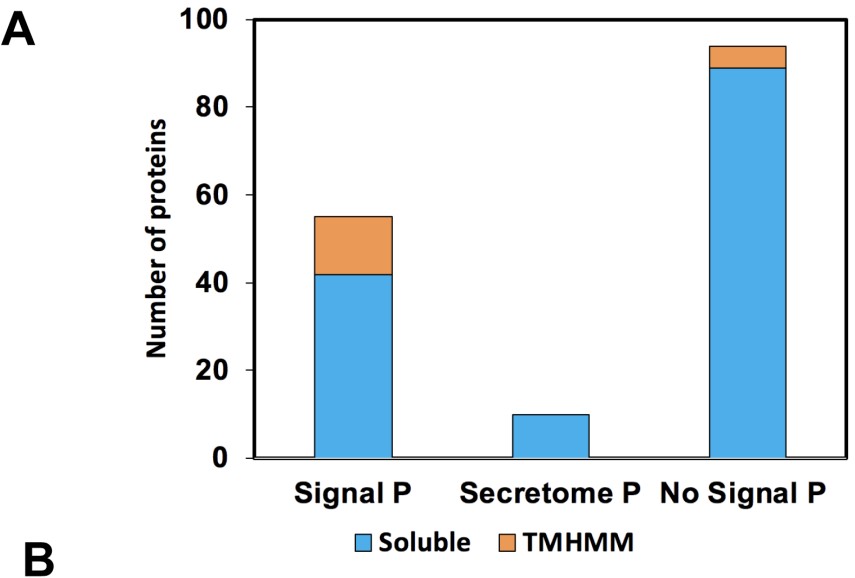

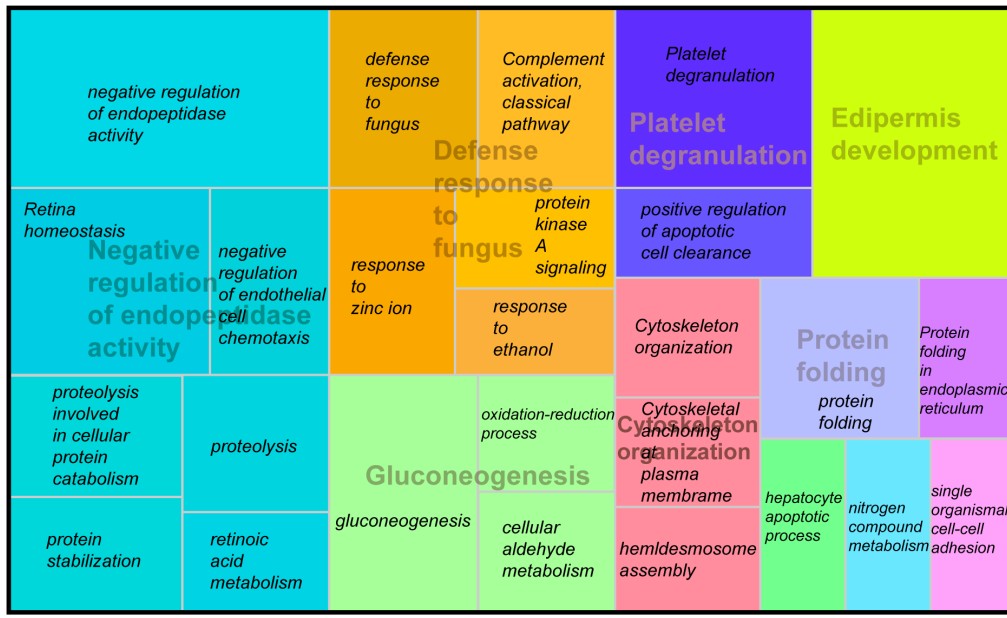

**Figure 2** **Classification of proteins identified in black howler monkey (*Alouatta pigra*) saliva by Nano LC-MS/MS.** (**A**) Prediction of N-terminal endoplasmic reticulum (ER) targeting signal peptide (SP, http://www.cbs.dtu.dk/services/SignalP/, *Petersen et al., 2011*), non-classical secretion (http://www.cbs.dtu.dk/services/SecretomeP/, *Bendtsen et al., 2004*), and transmembrane helices (TMHMM, http://www.cbs.dtu.dk/services/TMHMM/) in identified proteins. (**B**) Proteins were classified base GO ontology enrichment using David ontology tool (https://david.ncifcrf.gov/). We used REVIGO web server (http://revigo.irb.hr/) for the visual representation of the clustering of biological process. Names in italics indicate the GO enrichment of biological process and names with transparency indicate the clusters obtained by REVIGO using abs_log10_pvalue.

(GO:0050821), retina homeostasis (GO:0001895), retinoic acid metabolism (GO:0042573), negative regulation of endopeptidase activity (GO:0010951), and negative regulation of endothelial cell chemotaxis (GO:2001027).

The second most prominent cluster was the defense response to fungus conglomerating GO like protein kinase A signaling (GO:0010737), complement activation classical pathway GO:0006958 (GO:0006958), defense response to fungus (GO:0050832), response to ethanol (GO:0045471), and zinc ion (GO:0010043). The third most representative cluster named gluconeogenesis gathered the GO oxidation–reduction process (GO:0055114), cellular aldehyde metabolism (GO:0006081), and gluconeogenesis (GO:0006094).

### Howler monkey salivary proteins associated with host-defense in mammals

It is widely recognized that salivary proteins have many functional properties, and some have more than one function. According to data available on UniProt functional annotation (http://www.uniprot.org/) and review papers on salivary proteomics/peptidomics from humans and other mammals, we identified 10 proteins with dual function, including oral food perception and host-defense (6.4% of total identified proteins). We also identified proteins related with taste sensitivity or innate/acquired immunity (Fig. 3). We identified 28 salivary proteins/peptides (17.9% of total identified proteins) associated with functions, such as host defense, innate immunity, and antimicrobial properties (anti-bacterial, antiviral and anti-fungal). There were identified cationic peptides, and defense proteins (such as immunoglobulins) that have been reported as effective against parasites, fungi and cancer cells. Table 1 presents the complete list of proteins/peptides identified in saliva of howler monkeys related with host-defense and anti-microbial properties, and the references where the link between these proteins and that immune function has been reported.

### Howler monkey salivary proteins associated with oral food perception

We detected 16 proteins in saliva of howler monkeys (10.25% of total identified proteins) related with oral food perception; the complete list is shown in Table 2. There were identified six proteins associated with gustatory sensitivity of sweet, salty, umami, fatty-acids, and pungent flavors. For instance, carbonic anhydrase VI or "gustin" was identified and plays an important role in human taste perception of bitterness or fatty acids (*Morzel et al., 2014*). Likewise we identified four types of cystatins, histidine-rich glycoprotein, and IgA, which are associated with a major inhibition of the feeling of astringency and bitter taste (*Nayak & Carpenter, 2008*; *Canon & Neyraud, 2017*; *Shimada, 2006*).

## DISCUSSION

We identified 156 salivary proteins from black howler monkey (*Alouatta pigra*); a leaf and fruit eating primate that belongs to the most folivorous New World primate genus. The distinct proteins identified belong to most protein families described in mammals (*De Sousa-Pereira et al., 2015*); we categorized them according their likely function based

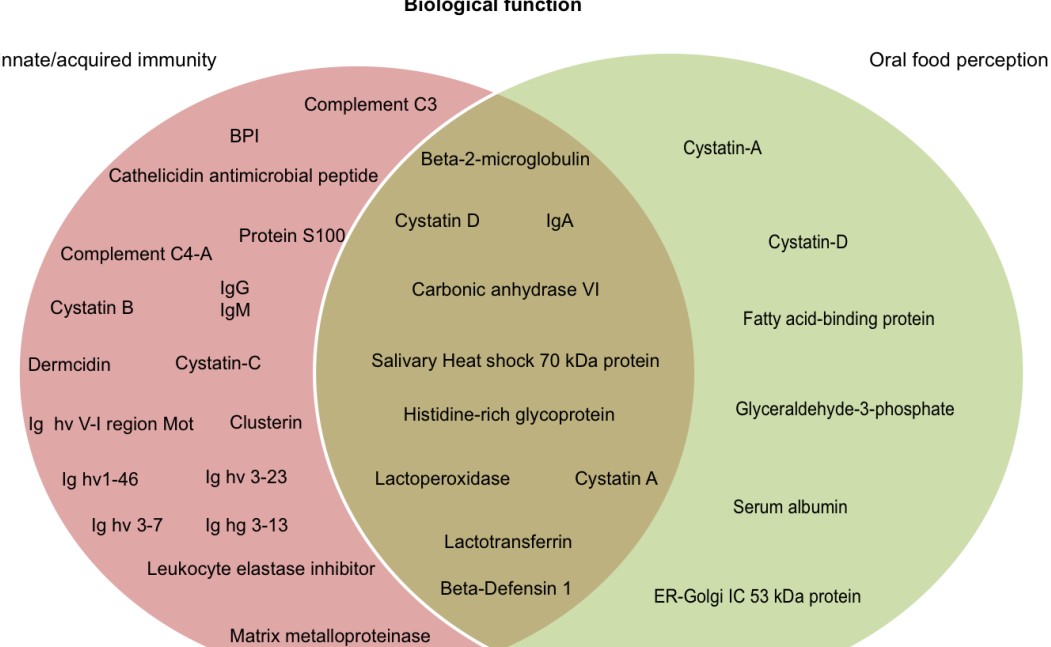

**Biological function**

Innate/acquired immunity | Oral food perception

Complement C3
BPI
Cathelicidin antimicrobial peptide
Beta-2-microglobulin
Cystatin-A
Protein S100
Cystatin D          IgA
Complement C4-A
Cystatin-D
IgG
Cystatin B          IgM
Carbonic anhydrase VI
Fatty acid-binding protein
Dermcidin          Cystatin-C
Salivary Heat shock 70 kDa protein
Glyceraldehyde-3-phosphate
Ig hv V-I region Mot          Clusterin
Histidine-rich glycoprotein
Ig hv1-46          Ig hv 3-23
Serum albumin
Ig hv 3-7          Ig hg 3-13
Lactoperoxidase          Cystatin A
Leukocyte elastase inhibitor
Lactotransferrin
ER-Golgi IC 53 kDa protein
Beta-Defensin 1
Matrix metalloproteinase

**Figure 3** **Probable biological role of the salivary proteins of howler monkeys.** Proteins are grouped according their participation in immunity and oral food perception reported in UniProt functional annotation (http://www.uniprot.org/) and in papers on salivary proteomics/peptidomics from humans and other animals. Ten proteins are involved in both biological functions.

on previous literature, nevertheless, we only can speculate about the function of these salivary proteins related to host defense and oral food perception in howler monkeys. Some proteins we identified have dual functions in oral food perception and innate immunity, which molecular weights correspond to the protein bands with higher densities in 1D-SDS PAGE (10–17 kDa) as cystatins and histidine-rich glycoprotein. This may indicate they are secreted in higher concentrations in saliva of howlers; however, their functional importance in howler monkey saliva remains to be investigated. We found by LC-MS/MS for the first time in saliva of primates, three types of salivary cystatins (A, B, and D); for instance, in humans have been described three S-type cystatins and C-cystatin (*De Sousa-Pereira et al., 2015*; *Vitorino et al., 2004*), also only S-type cystatins have been found in apes as western lowland gorillas (*Gorilla gorilla*) and chimpanzees (*Pan troglodytes*) (*Thamadilok et al., 2019*). Our results emphasize the essential physiological role that salivary proteins may have in maintaining the host-defense capacity and evaluating food properties, including taste and astringency. To the best of our knowledge, our study provides the first evaluation of the salivary proteome of a wild Neotropical primate. We provide a high number of predicted intracellular proteins—up to 57.05% of total identified proteins. Some proteins were predicted to have non-classical secretion (Fig. 2A, Table S1), thus, further experimental validation of their subcellular location is needed.

**Table 1** Salivary proteins associated with host defense of mammals, identified in the saliva of the Neotropical black howler monkey *Alouatta pigra* by Nano LC-MS/MS.

| Protein (Uniprot accession number) | Peptide sequence | MASCOT score | Function | Reference |
|---|---|---|---|---|
| Beta-2-microglobulin | EVDEQMLNVVNK | 38.5 | Immune response, involved in the presentation of peptide antigens to the immune system. Component of the class I major histocompatibility complex. | *Li, Dong & Wang (2016)* |
| Bactericidal permeability-increasing protein BPI (Q8TDL5) | VINEPTAAAMAYGLHK | 245 | Innate immunity in mouth, nose and lungs; binds bacterial lipopolysaccharide, bactericidal against both smooth and rough forms of Gram-negative bacteria, including *Neisseria meningitides* | *Bingle & Craven (2004)*, *Marra et al. (1990)* and *Shin et al. (2011)* |
| Beta-Defensin 1 (Q95M66) | MLMLAAQNILNPKDGKPVV TPSQDMVLGNYYLTMEEEGR | 51 | Antibacterial, antiviral and antifungal activity. Defense response to Gram-negative and Gram-positive bacterium, important antimicrobial effect against mycobactaria | *Wiesner & Vilcinskas (2010)* |
| Carbonic anhydrase VI (P23280) | HVIEIHIVHYNSK | 35.345 | Anti-caries protein in saliva | *Kimoto et al. (2006)* |
| Cathelicidin antimicrobial peptide (Q1KLX0) | LTALGQLLR | 47.03 | Antimicrobial activity against gram-negative and gram-positive bacteria and *Candida albicans* and are effective in vitro against oral microorganisms such as *Streptococcus mutans*, *Porphyromonas gingivalis*, and *Actinobacillus actinomycetemcomitans* | *Tao et al. (2005)* and *Dale & Fredericks (2004)* |
| Clusterin (P10909) | AATESFASDPILYRPVAVA LDTKGPEIR | 34.44 | Antimicrobial humoral response, innate immunity, response to virus | Uniprot.orross; *Amerongen & Veerman (2002)* |
| Complement C3 (P01024) | SLGLNPNHIHIYSASWGPEDDGK | 150.64 | Plays a central role in the activation of the complement system. Immune and inflammatory response. | *Ross & Densen (1984)* |
| Complement C4-A (P0C0L4) | TLVTQNSGVEALIHAILR | 117.4 | Innate immune and inflammatory response. Complement activation, classical pathway. | *Ross & Densen (1984)* |
| Cystatin B (Q8I030) | SCHLAMAPNHAVVSR | 196.35 | Innate immunity, inhibit proteases of bacteria. | *Fábián et al. (2012)*, *Dsamou et al., 2012*, *Blaydon et al. (2011)* and *Henskens, Veerman & Nieuw Amerongen (1996)* |

**Table 1** (*continued*)

| Protein (Uniprot accession number) | Peptide sequence | MASCOT score | Function | Reference |
|---|---|---|---|---|
| Cystatin-A (P01040) | GQPFEVLIIASDDGFK | 60.9 | Innate immunity, inhibit proteases, favor cell–cell adhesion. Is able to protect skin barrier from allergic reactions, including atopic dermatitis. Inhibition proteolytic activity of major allergens | *Fábián et al. (2012), Magister & Kos (2013)* and *Blaydon et al. (2011)* |
| Cystatin-C (O19093) | ALEEANADLEV, VLDELTLAR, APSTYGGGLSVSSSR | 95.95 | Found in high concentrations in body fluids. Promiment in immune cells. Strong inhibitor of all papain-like proteases. | *Magister & Kos (2013)* |
| Cystatin-D (P28325) | LGDSWDVK | 79.62 | Has a function in saliva as inhibitor of either endogenous or exogenous enzymes with cathepsin S- or H-like properties, inhibit proteases of bacteria | *Balbin et al. (1994)* |
| Dermcidin (P81605) | VTSFLDPWADPFGSGYQLTQS LMAFGRGGFFGQGLGNSVQK | 58.41 | Antimicrobial activity thereby limiting skin infection by potential pathogens in the first few hours after bacterial colonization. Highly effective against *E.coli, E.faecalis, S.aureus* and *C.albicans* | *Schittek (2012)* |
| Histidine-rich glycoprotein (P04196) | GTFAQLSELHCDKLHVDPENF, VLGAFSDGLAHLDNLK, VNVDEVGGEALGR, KVLGAFSDGLAHLDNLK, LLGNVLVCVLAQHFGK | 149 | Antimicrobial humoral immune response mediated by antimicrobial peptide. Antibacterial, antiviral and antifungal activity, overall against *C. albicans, Trichosporon pullulans* and *Cryptococcus neoformans*. Chemotaxis | *Wiesner & Vilcinskas (2010), Troxler et al. (1990), Oppenheim et al. (1988)* and *Jensen et al. (1994)* |
| Ig heavy chain V-I region Mot (P06326) | QVQLVQSGAEVK | 52.43 | V region of the variable domain of immunoglobulin heavy chains that participates in the antigen recognition. Humoral immunity | *McHeyzer-Williams et al. (2012)* |
| Immunoglobulin heavy constant alpha 1, IgA (P01876) | WLQGSQELPR, GFSPKDVLVR | 70.03 | More abundant in whole saliva. Protects mucosal surfaces from toxins, viruses, and bacteria by means of direct neutralization or prevention of binding to the mucosal surface | *Schroeder & Cavacini (2010)* and *Teeuw et al. (2004)* |
| Immunoglobulin heavy constant gamma 1, IgG (P01857) | WQQGNVFSCSVMH EALHNHYTQK | 60.58 | Immune response, including neutralization of toxins and viruses. Predominant isotype found in the body. It has the longest serum half-life of all immunoglobulin isotypes | *Schroeder & Cavacini (2010)* |

**Table 1** (*continued*)

| Protein (Uniprot accession number) | Peptide sequence | MASCOT score | Function | Reference |
|---|---|---|---|---|
| Immunoglobulin heavy constant mu, IgM (P01871) | LICQATGFSPR, VFAIPPSFASIFLTK | 49.276 | Adaptive immune response, antibacterial humoral response. Inactivate parasites, bacteria, and fungi | *Biesbrock, Reddy & Levine (1991)*, *Mehrotra, Thorton & Sheehan (1998)*; *Dsamou et al. (2012)*, *Mounayar et al. (2014)*, *Geisberger, Lamers & Achatz (2006)* and *McHeyzer-Williams et al. (2012)* |
| Immunoglobulin heavy variable 1-46 (P01743) | SEDTAVYYCAR | 40.76 | V region of the variable domain of immunoglobulin heavy chains that participates in the antigen recognition. Humoral immunity. | *McHeyzer-Williams et al. (2012)* |
| Immunoglobulin heavy variable 3-13 (P01766) | EVQLVESGGGLVQPGGSLR | 49.65 | Antimicrobial humoral immune; defense response to bacterium. | *Schroeder & Cavacini (2010)* |
| Immunoglobulin heavy variable 3-23 (P01764) | AEDTAVYYCAK | 45.47 | Antimicrobial humoral immune; defense response to bacterium. | *Schroeder & Cavacini (2010)* |
| Immunoglobulin heavy variable 3-7 (P01781) | NSLYLQMNSLR | 51.09 | Antigen binding. Humoral immunity | *Schroeder & Cavacini (2010)* |
| Leukocyte elastase inhibitor (P30740) | HNSSGSILFLGR | 58.3 | Anti-inflamatory | *Doumas, Kolokotronis & Stefanopoulos (2005)* |
| Lactoperoxidase LPO (P22079) | GSYNPVTHIYTAQDVK | 478 | Defense response to bacterium. Effective against *Pseudomonas aeruginosa, Burkholderia cepacia* and *Haemophilus influenzae* | *Thomas et al. (1994)* and *Wijkstrom-Frei et al. (2003)* |
| Lactotransferrin (P02788) | GFFEVTHDVSQLTCADFLR | 335 | Bacteriostatic, microbicidic, action against parasites. Prevent bacterial biofilm development in *P. aeruginosa* infection. Antifungal activity against *C.albicans* | *Groenink et al. (1999)*, *Lupetti et al. (2000)* and *Wiesner & Vilcinskas (2010)* |
| Matrix metalloproteinase (F6W5A7) | AFALWSAVTPLTFTR | 35 | Inhibitor of metallo-proteinases. Leukocyte migration | *Hayakawa et al. (1994)* |
| Protein S100-A8 (P05109) | AQEILSQLPIK | 97 | Acute inflammatory response; Plays a prominent role in the regulation of inflammatory processes and immune response. Induce neutrophil chemotaxis and adhesión. Defense response to bacteria, fungus. | *Lorenz et al. (2008)* and *Nacken et al. (2003)* |
| Salivary Heat shock 70 kDa protein (Q5R7D3) | RPTELLSNPQFIVDGATR | 259 | Binding of bacteria, immune response | *Fábián et al. (2009)* |

Espinosa-Gómez et al. (2020), *PeerJ*, DOI 10.7717/peerj.9489

**Table 2  Proteins associated with oral food perception identified in saliva of the Neotropical black howler monkey *Alouatta pigra* by Nano LC-MS/MS.**

| Protein (Uniprot accession number) | Peptide sequence | MASCOT score | Function | Reference |
|---|---|---|---|---|
| Beta-2-microbulin (O77523) Beta-Defensin 1 (Q9QWJ9 Q95M66) | EVDEQMLNVVNK | 38.53 | Reduce gustatory sense of sour flavors | *Neyraud et al. (2006)* |
| | MLMLAAQNILNPKDGKPVV TPSQDMVLGNYYLTMEEEGR | 51 | Gustatory sense of salty flavors | *Silletti, Bult & Stieger (2012)* |
| Carbonic anhydrase VI (Q9QWJ9 P23280) | HVIEIHIVHYNSK | 35.345 | Higher concentrations are related to lower acceptance of bitter solutions. Positivity related to taste sensitivity of fatty acids. Related with pungent flavors. | *Morzel et al. (2014)*, *Mounayar et al. (2014)* and *Canon & Neyraud (2017)* |
| Cystatin-A (Q9QWJ9 P01040) | GQPFEVLIIASDDGFK | 60.9 | Lower levels of Cystatins are related to hypersensitivity of astringency and bitter taste | *Dsamou et al. (2012)*, *Dinnella et al. (2010)* and *Morzel et al. (2014)* |
| Cystatin-D (Q9QWJ9 P28325) | LGDSWDVK | 79.62 | Positivity related to taste sensitivity of fatty acids. Reduce hypersensitivity to bitterness | *Mounayar et al. (2014)* |
| ER-Golgi intermediate compartment 53 kDa protein (F6SS58) | IDNSQVESGSLEDDWDFLPPKK | 57.322 | Mannose binding, sweet taste | Uniprot.org |
| Fatty acid-binding protein (Q01469) | LEDEIDFLAQELAR | 92 | Fatty-acid taste. High specificity for fatty acids, lipid binding | *Mounayar et al. (2014)* |
| Glyceraldehyde-3-phosphate (F7HS59) | HVVYPTAWMNQLPLLAAIEIQK | 28.69 | Reduce sensitivity of bitter taste | *Quintana et al. (2009)* |
| Histidine-rich glycoprotein (P04196) | GTFAQLSELHCDKLHVDPENF, VLGAFSDGLAHLDNLK, VNVDEVGGEALGR, KVLGAFSDGLAHLDNLK, LLGNVLVCVLAQHFGK | 149 | Are involved in the sensation of astringency, can decrease astringent sensation. Tannin-binding salivary proteins; play protective role to the pellicle by the scavenging tannins | *Dinnella et al. (2010)*, *Wiesner & Vilcinskas (2010)*, *Troxler et al. (1990)* and *Oppenheim et al. (1988)*. |
| Immunoglobulin, IgA (P01876) | WLQGSQELPR, GFSPKDVLVR | 70.03 | Higher concentrations are related to hypersensitivity of bitter taste. Positivity related to taste sensitivity of fatty acids. | *Dsamou et al. (2012)* and *Mounayar et al. (2014)* |

Espinosa-Gómez et al. (2020), *PeerJ*, DOI 10.7717/peerj.9489

**Table 2** (*continued*)

| Protein (Uniprot accession number) | Peptide sequence | MASCOT score | Function | Reference |
|---|---|---|---|---|
| Lactoperoxidase LPO (P22079) | GSYNPVTHIYTAQDVK | 478 | Reduce hypersensitivity to bitterness | *Morzel et al. (2014)* and *Fábián et al. (2015)* |
| Lactotransferrin (P02788) | GFFEVTHDVSQLTCADFLR | 335 | Sweet | *Becerra et al. (2003)* |
| Salivary Heat shock 70 kDa protein (Q5R7D3) | RPTELLSNPQFIVDGATR | 259 | Related to umami taste or glutamate taste sensitivity. Reduce sensitivity of pungent flavors. | *Fábián et al. (2015)* and *Canon & Neyraud (2017)* |
| Serum albumin (F7HCH2) | NVIPALELVEPIKK | 68.829 | Higher concentrations are related to hypersensitivity of bitter taste | *Quintana et al. (2009)* and *Dsamou et al. (2012)* |

## Salivary proteins linked with host-defense in mammals

A major finding of our research with howler monkeys is the identification salivary proteins and cationic molecules belonging to the two major antimicrobial peptides families: cathelicidins and defensins that rapidly inactivate infectious agents (*Wiesner & Vilcinskas, 2010*; *Zanetti, 2005*). Cathelicidins have been identified in cattle, sheep, rat, and dogs, but not in humans (*De Sousa-Pereira et al., 2015*). We also identified the antimicrobial peptide dermcidin that is recognized as a first line of skin defense in primates and has been identified in eccrine sweat glands of humans. Some argue that dermcidin is not found in other body fluids, such as nasal secretions, tears, saliva, semen, milk, and urine (*Schittek, 2012*); however, we identified this peptide in saliva of howler monkeys and it has been found in tears and cervicovaginal fluid in humans (*Shaw, Smith & Diamandis, 2007*). Dermcidin-homologous genes exist only in apes (*Pan troglodytes*, *Gorilla gorilla*, *Pongo abelii*) and Old and New World monkeys (*Schittek, 2012*).

Our proteomic analysis identified four members of the cystatin family (A, B, C, and D) in saliva of howlers that may inhibit the action of endogenous, bacterial, and parasitic protozoan proteases (*Fábián et al., 2012*). Similarly, the GO analysis of the salivary proteins indicates the most representative group corresponded to negative regulation of endopeptidase activity (Fig. 2B). Cystatins comprise a large superfamily of related proteins with diverse biological activities found in variable tissues, but salivary cystatins are important due their functions in immunimodulation, antimicrobial, and antiviral (*Dickinson, 2002*). A number of members of this protein family have been identified in saliva of humans (*Carneiro et al., 2012*) and in different mammals (e.g., cystatin D has been found in rat, cystatin S in dogs, cystatin C is present in Artiodactyla, Rodentia, Lagomorpha, Carnivora, and Primates (*De Sousa-Pereira et al., 2015*).

As one would expect, we identified carbonic anhydrase VI (CA-VI), which is an active mammalian isozyme specifically secreted by salivary glands that have multiple functions (*Kivelä et al., 1999*). The CA-family are zinc metalloenzymes responsible for the conversion of carbon dioxide to bicarbonate ($CO_2 + H_2O \leftrightarrow HCO_3$), which buffers saliva. CA-VI also has the ability to bind enamel and act in pH homeostasis of oral cavity and prevention of dental caries (*Kimoto et al., 2006*). Adding strength to the host-defense capacity of *Alouatta pigra*, we identified lactoperoxidase LPO, bactericidal permeability-increasing protein BPI, and histidine-rich glycoprotein, that are primarily responsible for innate immunity (*Bingle & Craven, 2004*; *Marra et al., 1990*; *Shin et al., 2011*; *Wiesner & Vilcinskas, 2010*; *Wijkstrom-Frei et al., 2003*). The microglobulin we identified is critical for immune modulation in vertebrate animals and has been identified as a biomarker for cancer cells malignancies (*Li, Dong & Wang, 2016*). Salivary heat shock 70 kDa protein may represent an important immune defense mechanism in saliva of howlers, as this protein has been identified in humans to bind bacteria and increases the release of proinflammatory cytokines from immune cells (*Fábián et al., 2012*; *Fábián et al., 2009*). It is important to emphasize the presence of three salivary secretory immunoglobulins in saliva of howlers as IgA, IgG, IgM and other five isoforms (Table 1). IgA is known to induce an antigen-unspecific manner by commensal microbiota; therefore, these secretory antibodies may bind multiple antigens

and are thought to eliminate commensal bacteria and self-antigens to avoid systemic recognition (*Schroeder & Cavacini, 2010*; *Teeuw et al., 2004*).

Several salivary proteins related with innate immunity of mammals were not identified in black howler monkeys (e.g., mucins *De Sousa-Pereira et al., 2015*). The failure to detect mucins may be due to the difficulty of assessing these proteins because of their large molecular mass, high viscosity, and poor solubility in aqueous solvents (*Herzberg et al., 1979*). However, we recognized that our preparation procedure of saliva samples, using 16,000 g × 10 min to separate the supernatant could result in loss of mucins in the precipitate. Other important proteins highly related to oral homeostasis, such as sthaterins and PRPs were also not identified.

We actually observed pink-staining bands in our SDS-PAGE gels, following the procedures suggested by *Beeley et al. (1991)* to detect PRPs, suggesting the presence of these proteins; however, some factors in our method could have interfered to detect PRPs such as the centrifugation process, and the use of trypsin for the protein digestion. Moreover, it is known that the identification of PRPs by mass spectrometry is unusually difficult (*Leymarie et al., 2002*); it could be possible that multiple PTM generated specific mass spectrum of modified peptides, which mass/charge (m/z) values could match with public databases (*Kim, Zhong & Pandey, 2016*). It is also possible that the sequences of PRPs are highly specific in *Alouatta pigra*.

Many of the proteins identified in howler monkey saliva are likely components of the early mammalian host defense against infection (*De Sousa-Pereira et al., 2015*). However, howler monkey saliva may have evolved a specific set of protein families to help them cope with infection risk and permit them to deal with habitat loss, fragmentation, and nutritional stress. (*Chapman, Gillespie & Goldberg, 2005*; *Chapman et al., 2013*). Zoonotic protozoa infection is related to degree of human contact with wild howler monkeys (*Kowalewski et al., 2011*).

## Salivary proteins linked with taste perception and food preference

We identified several important proteins in saliva of howler monkeys that might allow them to be selective and discerning while feeding, likely facilitate their feeding selectivity. Salivary proteins to perceive beneficial traits of food were found (e.g., CA-VI, lactotransferrin, ER-Golgi intermediate compartment 53 kDa protein, microbulin, defensing, cystatin D, fatty acid-binding protein, salivary heat shock 70 kDa protein, and IgA). We also identified salivary proteins that have been related to acceptance/detection of bitter and astringent solutions in humans, which may help howler monkeys to perceive and cope with negative characteristics of food, such as bitterness and astringency (related to plant secondary metabolites and toxic compounds) including cystatins (*Dsamou et al., 2012*; *Morzel et al., 2014*; *Mounayar et al., 2014*; *Quintana et al., 2009*), histidine-rich glycoprotein (*Dinnella et al., 2010*), albumin and IgA (*Dsamou et al., 2012*). The feeding flexibility of howler monkeys enables them to thrive in small and disturbed habitat patches, where food scarcity is common (*Chaves & Bicca-Marques, 2016*). To our knowledge, physiological studies on taste in howler monkeys have not been conducted and there are no data of taste detection thresholds or on the ability to discriminate between different qualities of tastants (*Salazar,*

*Dominy & Laska, 2015*). Consequently, for now we can only assume that the salivary proteins that we identified help these primates to choose the right diet.

Humans can differentiate among five flavors: sweet, sour, salty, bitter and umami (*Van Dongen et al., 2012*); although recently it has been proposed that humans can taste fatty acids (*Mattes, 2011*). Generally, it is accepted that each taste quality in food is related to its nutritional content (e.g., sweetness is associated with sugar, mono, and disaccharides; saltiness with sodium and protein content (*Van Dongen et al., 2012*); and umami with sodium and protein (*Van Langeveld et al., 2017*). Also, gustatory stimuli categorized as bitter and sour are associated with compounds that are potentially harmful (e.g., free protons or organic acid; bitter taste is related some toxins, *Lamy et al., 2016*).

Howler monkeys are herbivorous energy-maximizers and their diet is mainly leaves and ripe/unripe fruits (*Chapman, 1987*; *Chapman, 1988*; *Righini, Garber & Rothman, 2017*), but they can feed only on leaves for extended periods (*Behie & Pavelka, 2005*). A fruit-based diet is linked with a low protein intake and a decrease in mineral concentration (*Silver et al., 2000*), which may require selecting protein-rich and mineral-rich food items. For this purpose, howlers may benefit by secreting salivary proteins associated with gustatory sensitivity of salty and umami flavors (e.g., beta-defensin, CA-VI; cystatin D, and fatty acid-binding protein, IgA, salivary heat shock 70 kDa protein, Table 2). A leaf-diet is a diet poor in energy and fatty-acids, but high in fiber and often tannins (*Righini, Garber & Rothman, 2017*; *Espinosa-Gómez et al., 2018*), which makes selecting food difficult (*Silver et al., 2000*). Under these conditions, monkeys should select food items high in energy (carbohydrates, fatty-acids), but low in PSMs (tannins). Some studies in humans have found a relationship between sweet taste sensitivity and salivary proteins as cystatins (*Rodrigues et al., 2019*) and CA-VI (*Rodrigues et al., 2017*). We found in howler monkeys saliva four varieties of cystatins (Tables 1 and 2), which may help them to increase their sensitivity for sweet foods, although it remains to be investigated. For fatty-acids or lipids, it has been shown that these nutrients are important in the diet of howler monkeys (*Righini, Garber & Rothman, 2017*) and free fatty acids are one of the most abundant classes of nutrient metabolites in black howler monkeys foods (*Amato et al., 2017*). CA-VI or "gustin" plays principal role in taste sensitivity of fatty acids and sweet, salty, and sour flavors (*Feeney & Hayes, 2014*).

Corresponding to howler monkeys' ability to feed on tannin-rich diet (*Espinosa-Gómez et al., 2015*; *Espinosa-Gómez et al., 2018*), we identified several salivary proteins that have been related with the capacity to accept astringent and bitter foods e.g., cystatins (*Dsamou et al., 2012*; *Dinnella et al., 2010*; *Quintana et al., 2009*), glyceraldehyde-3-phosphate (*Quintana et al., 2009*), lactoperoxidase (*Morzel et al., 2014*), histidine-rich glycoproteins (*Dinnella et al., 2010*), and albumin (*Dsamou et al., 2012*) (Table 2). PRPs, histatins, statherins, cystatins, and amylase are salivary proteins with considerable affinity for tannins and are involved in astringency and bitter taste (*Lamy et al., 2016*; *Torregrossa et al., 2014*). We did not identify the well-known salivary PRPs and statherins identified as first line of defense against tannins (*Shimada, 2006*). However, we observed in our electrophoresis gels strong bands with pink-staining that may indicate the presence of PRPs (*Beeley et al., 1991*). Similarly, mucins also seem to have a role in astringency, but we did not identify mucins.

This may be linked to their high molecular mass, high viscosity, and poor solubility in aqueous solvents (*Lamy et al., 2010*).

This study supports the suggestion that α-amylase is not a component of saliva of animals feeding only on plants due their low ingestion of starch (*Boehlke, Zierau & Hannig, 2015*), as this enzyme was not identified in saliva of howlers. Also, chitinase was not found in our proteomic analysis, which is consistent with howlers' feeding behavior as this protein has been identified in insectivorous-omnivorous non-human primates (*Tabata et al., 2019*).

## CONCLUSIONS

Our research characterized the salivary protein of wild black howler monkeys and for the first time used a proteomic approach. We identified salivary proteins involved in host defense and oral food perception that helps understand the ecological adaptability of this species. However, for now we can only speculate that their salivary protein array is an advantage to face infection risk and low quality diets present in disturbed habitats (*Chapman et al., 2013*; *Chapman, Gillespie & Goldberg, 2005*). Salivary protein composition correlates with the feeding behavior of herbivorous primary feeders with energy-maximizing strategy. We also identified several important proteins involved with detection of astringency and bitterness. Correspondingly to their low starch and invertebrates-free diet, we did not identify salivary amylase or chitinase. The identification of 28 proteins in saliva of howlers that have been described with anti-bacterial, anti-fungal, and anti-viral capacity, might be involved to facilitate this species' ecological adaptability.

## ACKNOWLEDGEMENTS

We thank Javier Hermida (DVM) for his professional support during the capture of the monkeys. We especially appreciate the work and dedication of field assistants Dolores Tejero, Antonio Jauregui, Monserrat Ayala, Celina Oliva, and Tonatiuh Fernando. We are also very grateful to José Miguel Elizalde-Contreras for his invaluable advice and work at the proteomic lab. The authors thank to Marcus Clauss for helpful comments on this manuscript.

### Funding

Fabiola Carolina Espinosa-Gómez was supported by a postdoctoral fellowship from National Council of Science and Technology, México CONACYT 232395 and 263706 and a grant for the acquisition of an Orbitrap Fusion Tribrid Mass spectrometer (U0004-2015-1, 259915). The funders had no role in study design, data collection and analysis, decision to publish, or preparation of the manuscript.

### Grant Disclosures

The following grant information was disclosed by the authors:
National Council of Science and Technology, México CONACYT: 232395, 263706.
Orbitrap Fusion Tribrid Mass spectrometer: U0004-2015-1, 259915.

## Competing Interests

The authors declare there are no competing interests.

## Author Contributions

- Fabiola Carolina Espinosa-Gómez and Eliel Ruíz-May conceived and designed the experiments, performed the experiments, analyzed the data, prepared figures and/or tables, authored or reviewed drafts of the paper, and approved the final draft.
- Juan Carlos Serio-Silva and Colin A. Chapman analyzed the data, authored or reviewed drafts of the paper, and approved the final draft.

## Animal Ethics

The following information was supplied relating to ethical approvals (i.e., approving body and any reference numbers):

All research protocols reported here were reviewed and approved by the government of Mexico (permit number: SEMARNAT SGPA/DGVS/10426/14). The handling techniques adhered to the guidelines and the legal requirements of Mexico. This research also adhered to the ARRIVE guidelines in the United Kingdom and we followed the 3R guidelines (i.e., refinement, replacement, reduction of animal use).

## Data Availability

The raw data from proteomics analysis are available in the Supplemental Files.

## Supplemental Information

Supplemental information for this article can be found online at http://dx.doi.org/10.7717/peerj.9489#supplemental-information.

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
