# Peer review of "Salivary proteome of a Neotropical primate: potential roles in host defense and oral food perception"

_PeerJ, doi:10.7717/peerj.9489_

## Round 0.1 · original submission · Major Revisions

The paper is deemed interesting and straightforward in its finding but both reviewers have noted basic reporting problems due to English language issues. PeerJ provides direction to resources for help with this and authors can also ask for help from English speaking colleagues. One reviewer has several suggestions for improving clarity in the methods and results sections and both suggest that the findings might be overstated for two reasons: 1) this is s descriptive study and 2) workers in the field are not entirely in agreement as to the basic nature of the salivary proteome.

In your response and revision, please pay careful attention to reviewer comments and demonstrate how you have addressed them or why you do not feel you need to address them in your response. Please also enlist aid in the correct use of English to ensure your paper communications clearly the scope of the study and the results.

Reviewer 1 ·

Basic reporting

The English needs a major edition
Sufficient literature background, being relevant for drawing the conclusions and discussing the results.
Figures and tables are ok.

Experimental design

It seems to be adequated, but some clarifications are needed, as I suggest in the comments to authors.
Technics abl to provide rigorous and relevant data

Validity of the findings

Valid, but the presentation and discussion should be revised, as suggested in the comments to the authors.

Additional comments

The manuscript presents interesting results, relating the saliva of a primate with diet. This was an interesting evolution from other articles where saliva of these animals was studied, but now adding information through proteomics approaches.
However, the manuscript has several major considerations that need to be addressed before it can be considered for publication. One of the major points is English, which needs to be edited, since it is difficulting the interpretation of some parts of the text. Moreover, introduction is too long, considering the type of information it contains. Some methodological aspects should also be added and results section reviewed. Concerning discussion, I add also some points that should be addressed, but some of them would be avoid if English was clear.
My point-to-point considerations:
Abstract
Line 39 - "and for oral food perception" instead of "and taste sensitivity". This because after the authors refer astringency, and this is not a taste sensation.
Line 54 - The lack of identification of proline-rich proteins may not mean a really lack in the presence of these proteins, but rather a failure in these to be identified by mass spectrometry following a digestion with trypsin. I would suggest not to highlight these proteins in abstract.

Introduction:
Line 62 - I don't think this is the right place to cite Shimada. This paper refers that some species have salivary tannin-binding proteins, which protect them from suffer the anti-nutritional and toxic effects of tannins. As the sentence is written it seems that this paper refers that saliva may help to recognize diet-toxins, in order to animals select... I would suggest to change the sentence, if the authors want to keep the reference.
Line 65 - "relates to" instead of "influence"

Line 78-81 - English needs to be edited. This is an example of sentence which concepts seem confuse, but this may be due to the English
Line 92 - What the authors really want to say with "over-accumulation"? Do they mean increased levels in response to astringency? Or higher levels in individuals responding with higher intensity to astringency? Probably this needs clarification.
Lines 95-98 - Another example of confusing sentence, that does not allow to understand well the concepts

Material and methods
Lines 154-155 - dimension of gels and % of acrilamide should be detailed.
Lines 157-158 - what was the composition of the CBBR-250 solution and the composition of the fixative solution used before staining? It is important to have this information, since these reagents have influence in to obtain, or not, the pink bands after destainning with acetic acid.

Results
Lines 256-257 - I would remove this sentence from this section, since thisis more discussion than results
Line 260 - The number of individuals for which these 13 protein bands were digested and subjected to LC MS-MS is not clear. Each of these 13 protein bands is a pol of the same band from all the animals, or did the authors run 13 bands * 14 individuals. This must be clarified.
Lines 317-332 - All this paragraph is more a discussion than results presentation. If, on one hand, I understand that the authors want to present these in results section, since they are listing the proteins identified, which are related with oral food perception, on the other hand, I think this is not the best way of presenting. I would suggest that the authors make a table where they can put a column for the proteins identifies, other column for the type of oral sensation that they relate with and other column with the references where the link between the slivary protein and that oral sensation has been reported.
Probably table 1 is already the type of table I a suggesting, so you can base the presentation of results in that and avoid all this description.

Discussion
Lines 340-341 - This sentence needs to be reviewed. As it is I cannot understand if the authors identified for the first time or if de Sousa-Parreira et al had identified before
Line 371 - Another sentence where correction is imperative to achieve rigour. As it is it appears that is specifically in the saliva of howlers that this protein has these functions. But the study tht is cited does not refer these animals.

Lines 381-382 - once more it appears that the cited study evaluated this for the howlers... Please correct the sentence
Lines 389-397 - Moreover, the preparation procedure, with centrifugation at 16000g and the analysis of supernatant result in loss of mucins in the precipitate
Lines 397-400 - I think the authors should not consider "strange" the absence of PRPs. The methodology used in this study does not allow us to expect to find these proteins. On one hand, the sample preparation, with centrifugation would remove potential PRPs complexes that could be formed by binding to food (polyphenols) constituents. Although no incubation was made in vitro, in the oral cavity saliva is in contact with food constituents and, in case of some degree of presence, it would be expected to be in the precipitate. Moreover, with digestion with trypsin it is not expected to identify PRPs by mass spectrometry easily, since they do not digest as it would be in theory, due to the prolines present. So, I think that the other types of possible explanations to the failure in identifying PRPs, or even the hypothesis of lacking in these animals makes not much sense.
Lines 413-416 - This sentence needs to be supported with references where these specific proteins have been observed to b linked with "negative" traits of foods, which I suppose the authors mean bitterness and astringency
Line 427 - I am not comfortable with saltiness being associated with fat and energy content... In humans, an indirect relationship with processed foods can be made, but in animals, with natural access to food it makes not so many sense. So, thinking in adaptative evolution of senses, I would avoid to make this relationship, at least without bibliographic support.

Line 441 - When the authors are stating that leafs have few energy, it makes no sense to me to refer that the presence of some salivary proteins would help to select the ones with more energy... I think the focus must b mainly concerning the need to avoid PSM. I am not stating that these proteins cannot help in other sensory processes, but since this manuscript does not have measured diet type, nor subjected the animals to different types of diets, the authors should focus with is most known from other studies and I think the potential relation between salivary proteins and PSMs is stronger that hypothesis with energy signaling in leafs.

Lines 444-445 - This hypothesis would benefit if supported for studies showing a relationship between cystatins and sweet food sensitivity

Lines 450 -451 - This was already stated in the previous paragraph. Please reorganize to have each of the different sensory/nutritional aspects of foods together
Lines 456-458 - This sentence, in this place, does not add value to the discussion, unless the authors refer that other defense mechanisms exist in humans (besides salivary proteins) and this can also be expected in monkeys. Even so, I think it is not clear to have this in sequence of the previous text.
Lines 459-462 - This sentence needs to be supported with references.

Line 467 - please remove this part of the sentence: “but did not identify these proteins”

Conclusions
Line 480 - As I referred before, I would prefer the authors to say "oral food perception" rather than "taste sensitivity", since they are discussing all the results with different types of sensations that can occur in mouth
Lines 486-487 - Since the approach used would not allow many conclusions about these proteins, I would opt by not referring them here or in abstract.

Reviewer 2 ·

Basic reporting

The paper was straight forward and easy to read for the most part. It could use the assistance of a language editor. Examples include but are not limited to line 63 "so that a nutritious non-toxins can be selected" and 66 "in a diversity of environment." Also there is some repetition eg. Lines 102-105 and 451-454 are almost identical.

Experimental design

This is not a hypothesis driven paper, it is descriptive in nature. Samples were collected from darted monkeys and analyzed.

Validity of the findings

My major concern with the paper is that the authors have defined and interpreted the proteins they found by categorizing them based on the literature which is at best unsure. For example, there is still quite a lot of debate about the role of some of these salivary proteins in taste and many are correlated with preference at best. I strongly urge the authors to temper their interpretations of these protein classes. For example, something as simple as acknowledging the uncertainty of the interpretation would help.
An example, although my concerns are not limited to line 409 should read ".. that may allow them to be selective and discerning while feeding"

I am also concerned by the lack of PRPs in the results. This suggests that a step in the identification of the proteins was not ideal for isolating and identifying the PRPs. It would be useful to have a bit more discussion about this issue.

Additional comments

This is a straightforward and well contained description of howler monkey saliva. I would caution the authors to tone down the interpretation of the proteins presence when the fields looking at those proteins are still in debate on the causal relationships between the proteins and the functions in many cases.

---

## Round 0.2 · accepted · Accept

The reviewers are both satisfied with the revised manuscript and recommended it be accepted. I concur. I have asked staff to make the small change suggested by Reviewer 1 in order to avoid you resubmitting the paper a third time. The change is minor and clarifies that bitter taste is also implicated along with fatty acids.

Reviewer 1 ·

Basic reporting

Corrected according suggestions.

Experimental design

OK

Validity of the findings

With the corrected version it is ok.

Additional comments

The authors were able to answer my concerns and the manuscript greatly improved. I consider it acceptable in its current form.
A very minor point, but that I think should be adressed is in the pargraph in lines 326 – 333, “human taste perception of fatty acids" can be replaced by “human taste perception of bitterness or fatty acids”.

Reviewer 2 ·

Basic reporting

Authors have addressed my concerns.

Experimental design

Clear and thoughtful

Validity of the findings

Underlying data are interesting and useful

Additional comments

The authors have addressed my concerns.